# Effects of an Innovative Introductory Course on the Professional Commitment of First-Year Undergraduate Nursing Students: A Quasi-Experimental Study

**DOI:** 10.3390/nursrep15090310

**Published:** 2025-08-25

**Authors:** Wenzhe Hua, Yinghui Wu, Yaru Tang, Daqiao Zhu, Qiong Fang

**Affiliations:** Shanghai Jiao Tong University School of Nursing, Shanghai 200025, China; huawenzhe1992@163.com (W.H.); yinghuiwu@shsmu.edu.cn (Y.W.); tangyaru0804@sjtu.edu.cn (Y.T.)

**Keywords:** introductory course, nursing student, pre–post study, professional commitment

## Abstract

**Aim:** We aimed to assess the impact of a Health and Nursing course on the professional commitment of first-year undergraduate nursing students and obtain quantitative and qualitative feedback on teaching. **Design:** This study employed a quasi-experimental design. Methods: Fifty first-year undergraduate nursing students who attended the Health and Nursing course were recruited. A self-designed basic information questionnaire, the Nursing Professional Commitment Scale, the Students’ Perceived Teaching Quality Questionnaire, and two quick open-ended questions were used to collect the data. Professional commitment before and after the intervention was compared using an independent samples *t*-test. The correlation between the students’ perceived teaching quality and professional commitment was assessed using Pearson’s correlation coefficient. Content analysis was used to analyze qualitative feedback. **Results:** The participants experienced a significant improvement in their professional commitment. The students’ perceived teaching quality was significantly correlated with their professional commitment. According to the students’ feedback, the most impressive aspects of the course were case-based learning and visits to healthcare institutions. **Conclusions:** A Health and Nursing course with high-quality teaching increased the professional commitment of first-year undergraduate nursing students. The findings suggest that nursing educators should consider the external macro-sociopolitical environment when designing an introductory course to equip students with a broader perspective on nursing professional development. Teaching content and pedagogy should be improved to promote knowledge delivery and internalization.

## 1. Introduction

Shortages in the nursing workforce plague the Chinese healthcare system and those of many other countries worldwide [1]. Approximately 31 million men and women constitute the global nursing and midwifery workforce [2]. Despite their large numbers, nurses and midwives represent more than 50% of the current shortage of healthcare workers [2]. One potential approach to addressing the shortage of healthcare professionals is to enhance professional commitment [3]. Professional commitment refers to an individual’s psychological response to their profession and is a known predictor of professional competence, job satisfaction, and turnover intention [4,5]. Undergraduate education is an important stage for nursing students to develop a professional commitment to their profession [6]. Enabling students to understand nurses’ contributions to the health field and build positive professional cognition as early as their first contact with the nursing profession is imperative for their professional commitment development and career decision-making [7,8].

To deepen professional commitment, it is essential for students to grasp the broader contexts in which nurses operate, including global health challenges and national health policies. Understanding these factors not only broadens perspectives but also reinforces recognition of nurses’ vital roles in healthcare and promotion, which strengthens their professional commitment. Global health is an essential competency for nurses, and its importance has continuously increased [9], especially after the COVID-19 pandemic. It is an area of study, research, and practice that prioritizes improving health and achieving health equity for all people worldwide [10]. Educators in the USA, Canada, and India have highlighted the importance of integrating global health concepts into nursing curricula [11,12,13]. However, most undergraduate nursing curricula lack a core course on global health [14,15]. Looking at global health challenges also involves addressing domestic health inequities through local policies and practices, as exemplified by initiatives such as Healthy China 2030. Launched in 2019, this important initiative aims to ensure that the Chinese population has access to health by advocating for the whole society’s participation in the concept of “Health for All, and All for Health” [16]. Nurses contribute to preventing diseases, assist in diagnosis and treatment, reduce pain and suffering, and promote rehabilitation, playing an essential role in implementing health policies and achieving public health goals [17]. As future nurses, nursing students should develop an understanding of global and national health policies and practices early in their studies, as this knowledge situates their personal career development within a broader international and national sociopolitical context and contributes to fostering their professional identity and commitment [11,15].

To respond to these needs, a Health and Nursing course was developed as an innovative introductory class in 2020, immediately after the launch of Healthy China 2030. This course aims to help nursing students perceive their professional environment and understand how policies are implemented in nursing practice. It is offered as the first specialized course in the first semester at university, serving as the initial step in shaping students’ understanding of the nursing major. In this course, nursing students learn about the concepts of global health and Healthy China 2030, the current conditions of the nursing profession, and the potential future nursing opportunities and challenges to renew their views of the profession based on their surroundings and social media. This course is part of a large teaching reform project that aims to improve the professional commitment of nursing students by redesigning and innovating nursing curricula.

### Background

Traditionally, the first specialized course for nursing students in China and other countries has been titled “Introduction to Nursing” or “Fundamentals of Nursing,” which covers the history, basic concepts, and fundamental knowledge and skills of nursing [18,19,20]. However, as the sociopolitical context of healthcare delivery changes, nurses must transform their roles and adapt to keep pace with such changes in society, as well as to respond to financial challenges to best manage resources and ensure safe, high-quality patient care [21]. Health policies reflect people’s health needs and provide a context for nurses’ practices, roles, and knowledge. Guiding nursing students to learn about general concepts and knowledge of health and become familiar with the external environment for health profession development is helpful in explaining specialized nursing concepts in subsequent courses. An innovative introductory course is needed to provide a broader perspective of the entire health field, rather than just the nursing profession, to equip future nurses with a strong sense of duty in health promotion.

The Health and Nursing course was developed based on the Nursing Metaparadigm, which identifies four major concepts that are frequently interrelated and fundamental to nursing theory: person, environment, health, and nursing [22]. The metaparadigm provides a theoretical foundation for the course’s framework [23]. The course content was framed around these four concepts and incorporated essential, advanced, and macro-knowledge of Health and Nursing (Figure 1). The course content echoes the aim of all nursing education, which is to ensure that graduates not only receive the necessary knowledge and skills to provide safe patient care but also establish a professional commitment to demonstrate continuous effort in nursing [24].

Professional commitment is a psychological reaction to a person’s profession that is initially shaped by basic education [25]. Students with higher professional commitment tend to exhibit higher professional commitment after graduation and are more willing to engage in nursing careers. Unfortunately, undergraduate nursing students in China have moderate levels of professional commitment [26,27]. According to research examining the factors influencing nursing students’ professional commitment, education plays a significant role [28]. The previous study conducted by the authors, which surveyed three educational institutions, suggested that the academic faculty’s beliefs about nursing were positively correlated with nursing students’ professional commitment. Other studies have suggested that students’ satisfaction with learning, teachers’ pedagogical skills, and the pedagogical atmosphere are positively related to perceived professional commitment [29,30].

Professional commitment in nursing is an ongoing, dynamic process that is shaped as students enter university. The first year is a crucial phase for correcting students’ previous cognitive biases, improving their value in the nursing profession, and building professional commitment. The literature and our previous study suggest that only a small percentage of students chose nursing majors because of their interest, whereas the majority chose them because of family suggestion, the good job prospects the nursing major offers, employment opportunity considerations, and professional transfer rather than their interests and ideals, resulting in low professional commitment [27,31]. In addition, their stereotypical images of nursing before they were enrolled in nursing schools were influenced by people who were unfamiliar with the profession [32]. Therefore, educators should pay more attention to cultivating professional commitment, clarifying misunderstandings, and promoting a conceptual and systematic understanding of nursing in students as early as their entry into university, particularly for those who are ambivalent about nursing as a career.

Most studies have focused on third- or fourth-year students and examined the effect of internship experience on their professional commitment [27,31]. These studies generally show a decline in professional commitment after internships, attributed to gaps between expectations and reality, unpleasant clinical experiences, and negative relationships with clinical instructors. However, fewer studies have investigated first-year students, even though this initial phase is crucial for shaping professional commitment. Research comparing students at different academic levels suggests that professional commitment tends to decrease as students advance to higher years [33]. For example, Kong et al. (2016) and Yu et al. (2023) found that the professional commitment of first-year students was higher than that of second-year students, which may be a result of increased academic pressure and decreased passion [34,35]. Therefore, nursing educators need to focus more on early-stage students by using innovative approaches to challenge preconceived notions and stereotypes. Providing an accurate and comprehensive image of the nursing profession that summarizes current trends and career opportunities can foster a stable and lasting professional commitment from the outset and counteract the decline seen in later years [36].

Some intervention studies have aimed to increase professional commitment by improving teaching content or procedures. Chang et al. (2024) reported that dialog with senior nursing students increased first-year nursing students’ professional commitment by helping them reflect on the role of nursing and establish professional values and responsibilities [37]. In another study, a one-week structured exemplar education program significantly improved the commitment of third-year students [26]. This research suggests that courses or programs aimed at increasing accurate and comprehensive awareness of the role and functions of professional nurses in promoting health could be effective in shaping students’ professional commitment.

The primary aim of this study was to examine whether the professional commitment of first-year undergraduate nursing students improved after taking the Health and Nursing course. The secondary aim was to evaluate the students’ perceived teaching quality and analyze qualitative feedback on this course.

## 2. Materials and Methods

### 2.1. Study Design

This is a quasi-experimental study without a control group, using pre- and post-intervention measurements.

### 2.2. Participants and Ethical Considerations

This study was set in a school of nursing at a top-ranking public university in Shanghai, a large metropolitan city in eastern China. Non-randomized convenience sampling was used to recruit participants. The students decided on the nursing major when they applied for the university through the national college entrance examination. The annual enrollment for the nursing school at this university is approximately 50 students due to its highly selective admission process. Because Health and Nursing is a compulsory course, all 50 of the first-year undergraduate nursing students who were required to take the course in the autumn quarter of 2023 were invited to participate in the pre- and post-tests. Before the pre-intervention test, students were informed of the purpose of the course and their freedom to withdraw from the study at any time. Informed consent was obtained from all participants. All surveys were anonymous, and the investigators could not see the students’ information. Ethical approval was obtained from the ethics committee of the university. To avoid any potential influence or concerns from students, none of the instructors teaching the Health and Nursing course was involved in the collection or analysis of research data. Participation was entirely voluntary, and no incentives or financial benefits were provided to the students. All 50 agreed to participate in this study and completed the pre-intervention survey. Two students who did not volunteer to complete the post-intervention survey were excluded.

### 2.3. The Health and Nursing Course

The Health and Nursing course included 34 class hours with two credits. The course lasted for eight weeks. The expected learning outcomes of the Health and Nursing course were as follows: develop an understanding of global and national health policies and practices; recognize the roles and contributions of nurses in healthcare delivery and health promotion; foster early professional identity and commitment to the nursing profession. These outcomes guided the course design and informed the evaluation of its impact on first-year nursing students. Four sequential modules were designed to gradually internalize the knowledge, skills, and values of the profession: expert lectures, visiting healthcare institutions, case-based learning (CBL), and team presentations (see Figure 2).

In Module One, seven professors with significant work experience from different disciplines, including nursing, public health, and public administration, gave lectures lasting three class hours on a specific topic. Seven topics were identified according to the Nursing Metaparadigm (Figure 1): global health and nursing, nursing’s role in promoting Healthy China, nursing discipline development, nursing humanities and One Health, health communication, multidisciplinary cooperation to promote health, and nursing research in promoting population health. Through this module of 21 class hours, the students learned about the basic knowledge of global health and health policies, core concepts in nursing practice, and advanced development in the nursing discipline.

In Module Two, two healthcare institutions were visited. One was a community-embedded elderly care institution, and the other was a proton therapy center. These two visits helped students understand the context of geriatric and oncological care, which are recognized as key areas of concern in public health. During the visits, students were guided by staff through the facilities and services; engaged in communication with patients; and held discussions with faculty regarding institutional services, patient care, and career development in specialized nursing fields. Through this module, students combined the theoretical knowledge they acquired in Module One with the practice context using a reflective process.

In Module Three, two separate CBL sessions were conducted, each lasting three class hours. In each session, students worked in teams to analyze and discuss a single thematic case scenario. CBL is an established approach used across disciplines in which students apply their knowledge to real-world scenarios, promoting higher levels of cognition [38,39]. The cases presented disciplinary problems for students to devise solutions using the knowledge they had learned in Modules One and Two. A total of 50 students were randomly grouped into four teams with 12 or 13 students per team. Each team was supervised in a single classroom by two instructors [38].

In Module Four, the students in each team worked together and created 15 min presentations. After completing the two CBL sessions, the four student teams drew lots to fairly determine which of the two case scenarios each team would present. Then, they had one week to prepare the presentation. They presented their understanding, analysis, and solutions regarding the problems in the case; their feelings about the health field or nursing profession; their ideas about their own career development; and their reflections during this course. Through this module, students cooperated in teams, communicated their thoughts across teams, and deeply reflected on the course.

### 2.4. Measures

#### 2.4.1. Participant Characteristics

A self-designed questionnaire was used to collect sociodemographic factors (gender, residence, and only-child status), academic-related factors (whether the participant was a student leader), and questions about the students’ and their families’ previous and current cognition and attitudes toward the nursing major. The survey items were determined based on our previous research on nursing students’ professional commitment, reviewing the relevant literature, and consulting with nursing education experts to ensure their validity and reliability.

#### 2.4.2. Professional Commitment

Professional commitment was evaluated using the self-reported Nursing Professional Commitment Scale developed by Lu et al. (2000) [40]. This instrument has been used in multiple nursing education studies to assess professional commitment among students (e.g., Wang et al., 2024; Kaya et al., 2022; Duran et al., 2021) [41,42,43]. It consists of 34 items with four dimensions: “willingness to make an effort” (16 items) evaluates the extent to which one individual is willing to exert considerable effort on behalf of the organization; “desire to stay in the profession” (8 items) reflects the degree to which a nursing student would like to remain in their current job; “intrinsic positive value of work” (5 items) reflects the individual’s perceived or calculated value of their current work; and “belief in goals and values” (5 items) evaluates a student’s belief in the goals and values of the nursing profession. All items were scored on a 4-point scale ranging from 1 (strongly disagree) to 4 (strongly agree), with higher scores indicating stronger professional commitment. The overall score for professional commitment is the sum of the scores of the four dimensions. Scores ranged from 34 to 136.

#### 2.4.3. Students’ Perceived Teaching Quality

The students’ perceived teaching quality was evaluated using a self-designed questionnaire. The questionnaire was newly developed for this study, based on a thorough literature review, the authors’ teaching experience, and consultation with experts in the field. A total of 20 items were used to assess students’ appraisal of the organization of the class, teaching attitudes, teaching skills, class content, course materials, and modalities. Each item was scored on a 4-point scale ranging from 1 (strongly disagree) to 4 (strongly agree), with higher scores indicating stronger teaching quality. The overall score for perceived teaching quality was the sum of all item scores and ranged from 20 to 80.

#### 2.4.4. Qualitative Feedback

Two quick open-ended questions were used to obtain the students’ qualitative feedback on this course: these were “What impresses you most in this course?” and “What can be improved through this course?” The questions were designed to complement the quantitative measures of perceived teaching qualities by allowing students to describe their impressions of the course, the teaching strategies, and the learning environment. This qualitative feedback can inform future improvements in teaching quality.

### 2.5. Data Collection

Participant characteristics (tool described in Section 2.4.1) and professional commitment (Section 2.4.2) were assessed as pre-test data one day before the first class. Post-test data collected three days after Module Four included participant characteristics (Section 2.4.1), professional commitment (Section 2.4.2), students’ perceived teaching quality (Section 2.4.3), and qualitative feedback from the two open-ended questions (Section 2.4.4). All data were collected via the Wenjuanxing online questionnaire system, and all items had to be completed before submission. The researchers responsible for the data collection and analysis were not connected to the “Health and Nursing” course but were actively involved in teaching other courses to nursing students, with over five years of nursing education experience. They were invited based on their familiarity with this study, expertise in undergraduate nursing curriculum, and teaching experience.

### 2.6. Data Analysis

This study was conducted using an anonymous survey design, which means that we could not match the pre-test and post-test responses to individual students. As a result, although two students did not participate in the post-test, we could not identify which specific students withdrew. Therefore, all 50 students were included in the pre-test analysis, and 48 students were included in the post-test analysis.

SPSS Statistics version 19.0 (IBM Corp., Armonk, NY, USA) was used to analyze the data. The Shapiro–Wilk test and Q-Q plot were used to assess the normality of professional commitment and students’ perceived teaching quality. The Shapiro–Wilk test indicated that the data were normally distributed (*p* > 0.05), which was consistent with the results of the Q-Q plots. Therefore, the mean and standard deviation were calculated to describe the distribution and central tendency of professional commitment and the students’ perceived teaching quality. Counts (n) and percentages (%) were used to describe the participants’ characteristics, which are independent variables. Qualitative feedback was analyzed using content analysis. A word cloud was generated based on the qualitative data to show the most frequently mentioned terms.

Professional commitment before and after the intervention was compared using an independent-sample *t*-test. To assess the magnitude of the course’s effect on professional commitment, Cohen’s d was calculated for all pre- and post-test comparisons. Pearson’s correlation analysis was used to examine whether there was a correlation between students’ perceived teaching quality and professional commitment. *p*-values < 0.05 were considered statistically significant.

## 3. Results

### 3.1. Participant Characteristics

All 50 students completed the pre-intervention survey, and 48 completed the post-intervention survey. The average time to complete the survey was approximately 6 min. The students’ basic characteristics are shown in Table 1. They and their families’ cognition and attitudes regarding the nursing major are shown in Table 2. Of the 50 students, the majority were female (36, 72.0%) and from urban areas (40, 80.0%). Thirteen (26.0%) students chose their major based on their own decisions, and eight (16.0%) were very interested in nursing when choosing a major. Approximately half the students (52.0%) knew well about their major when choosing it.

Two students did not respond to the post-intervention survey. After the course, the percentage of those who chose “very supportive” for their parents and siblings’ current attitudes toward their majoring in nursing increased from 26.0% to 41.7%, while those who chose “somewhat supportive” decreased from 46.0% to 31.3%. The percentage of students who knew a lot about their major upon completing the survey increased from 64.0% to 93.8%.

### 3.2. Professional Commitment Before and After the Course

In the present study, the Cronbach’s α value of the dimensions ranged from 0.711 to 0.981, indicating strong internal consistency. Before the course, the mean score for professional commitment was 93.00 (SD = 17.89), and the average score per item was 2.74 (SD = 0.53). Among the four dimensions, “belief in goals and values” scored the highest, while “willingness to make an effort” scored the lowest (Table 3). After the course, the mean score of professional commitment was 100.92 (SD = 20.27), and the average score per item was 2.97 (SD = 0.60). The “belief in goals and values” dimension scored the highest, while “desire to stay in the profession” scored the lowest.

The total score for professional commitment post-intervention was significantly higher than that of pre-intervention (*p* < 0.05, Cohen’s d = 0.41). Among the four dimensions, the differences in “willingness to make an effort” and “belief in goals and values” dimensions were significant (*p* < 0.05, Cohen’s d = 0.53), while the differences in “desire to stay in the profession” and “intrinsic positive value of work” dimensions were not (*p* > 0.05). The scores for professional commitment before and after the intervention are shown in detail in Table 3.

### 3.3. Students’ Perceived Teaching Quality

The Cronbach’s α value of the questionnaire was 0.965, indicating strong internal consistency. The mean total score for students’ perceived teaching quality was 73.94 (SD = 8.52), while the full score was 80. The mean item score ranged from 3.50 to 3.83 (Table 4). The items “The faculty respected students” and “The faculty responded to students’ comments and questions” scored the highest. The items “Fully online teaching can achieve the same learning outcomes as the current blended course format” and “Course content and materials could motivate learning enthusiasm and interest” scored the lowest (Table 4).

### 3.4. Correlation Between Professional Commitment and Students’ Perceived Teaching Quality

The total score for the students’ perceived teaching quality was positively correlated with the total score (r = 0.310, *p* = 0.032), as well as the “willingness to make an effort” (r = 0.380, *p* = 0.008) and “belief in goals and values” (r = 0.423, *p* = 0.003) dimensions of professional commitment. However, the other two dimensions were not (*p* > 0.05). The detailed results are shown in Table 5.

### 3.5. Qualitative Feedback on the Course

According to the students’ answers to the two open-ended questions, what most impressed them about the course were CBL and its group discussion form, visiting healthcare institutions, nursing humanities, and hospice care. Their suggestions for the course included the following: more visiting activities should be arranged; more credit hours for this course should be expected; CBL should be deeper; and student–faculty interactions, cases to help understand nursing knowledge, and student enthusiasm should be improved in lectures (see Figure 3).

## 4. Discussion

Using a quasi-experimental design, this study tested the effect of an innovative introductory course for undergraduate nursing students and found a significantly positive impact on students’ professional commitment. The students’ perceived teaching quality was positively correlated with their professional commitment. In addition, the students provided constructive feedback on their learning experiences by answering qualitative questions.

### 4.1. The Effect of the Course on Professional Commitment

The total score for professional commitment improved significantly after the Health and Nursing course, which is consistent with a previous study finding that introductory courses help with the decision to pursue nursing as a career [36,37]. Among the four dimensions, “willingness to make an effort” and “belief in goals and values” increased significantly, while “desire to stay in the profession” and “intrinsic positive value of work” did not. This suggests that, after the course, the students had a clearer understanding of the nursing profession and a greater intention to participate in future professional studies. This is consistent with a study suggesting that learning engagement and motivation are positively related to professional commitment [44]. The dimensions “desire to stay in the profession” and “intrinsic positive value of work” are about job and career commitment, which are used to evaluate if an individual is satisfied with their current work and willing to remain in that profession. The reason the two dimensions did not increase might be that first-year students were not deeply engaged in the profession and had no experience working in it. It is difficult for them to make a clear decision about their future work and to have a significant attitude change about this work during the initial stage of shaping professional commitments.

According to the participants’ characteristics, cognition, attitudes, and family members’ attitudes toward the nursing profession improved. This suggests that the course effectively corrected students’ and their families’ previous misunderstandings and highlighted interesting career paths in nursing. This was a new finding, as other studies have not examined changes in family attitudes. In fact, a family’s influence on a student’s attitude toward the profession could last until graduation and job-seeking [45]. The previous study (2022) of the authors’ team revealed that the family’s attitude toward nursing majors was positively associated with professional commitment at the time of graduation. According to the findings of this study, a student’s improved cognition and attitudes might affect their family, which needs to be explored in future studies.

### 4.2. Perceived Teaching Quality Level

The students’ perceived teaching quality was high, suggesting that they were mostly satisfied with the course and faculty. According to Table 3, “The faculty respected students,” “The faculty responded to students’ comments and questions,” “The faculty delivered positive professional values to students”, and “The faculty motivated and encouraged the students to think independently” scored the highest. These items suggest that faculty members had a positive attitude toward teaching and encouraged students’ critical thinking. It is useful to build a good student–instructor relationship, as this affects the quality of the teaching–learning process and plays a crucial role in shaping commitment through its impact on professional identity [46,47]. In addition, the results suggest that faculty members act as good role models with positive professional cognition. Social identity theory assumes that individuals are motivated to seek group membership to help define who they are and foster a positive self-image [48]. The faculty’s strong belief in the nursing profession is related to students’ professional commitment, as suggested by our previous study and others [49].

The items “Online learning would also achieve the expected effect” and “Course content and materials could motivate learning enthusiasm and interest” scored the lowest. The former suggests that a face-to-face format is irreplaceable. With the development of technology, online learning has become increasingly popular, particularly after COVID-19. However, in this course, the clinical visits and CBL were welcomed and highly appraised by the students; these offline teaching activities provided them with the opportunity to interact with healthcare staff, patients, and other students to build positive cognition of the nursing profession and strengthen professional commitment [20]. The latter item scored the highest, possibly because the course content is about the development of the Health and Nursing profession as it relates to sociopolitical contexts, which is somewhat boring and difficult for students to understand. This suggests that instructors should use cases or scenarios to stimulate interest and help students understand their course content. In addition, boredom can be decreased by a good learning environment and student–instructor interactions [50]. Combined with strategies such as humor, concern, and compassion, appropriate teaching pedagogy can lead to positive emotional results that increase student motivation and engagement and, in turn, professional commitment [51].

### 4.3. Correlation Between Perceived Teaching Quality and Professional Commitment

The total score for professional commitment and the dimensions “willingness to make an effort” and “belief in goals and values” were significantly correlated with the students’ perceived teaching quality, while the dimensions “desire to stay in the profession” and “intrinsic positive value of work” were not. These results are consistent with the *t*-test findings and may reflect both the timing of the intervention—early in the first semester—and the curriculum content, which focused on introducing the profession, promoting reflection, and fostering an understanding of professional roles. Consequently, it had more influence on “willingness to make an effort” and “belief in goals and values” than on longer-term career aspirations or intrinsic motivations. Previous studies focusing on senior students show that a good internship experience increases all dimensions of professional commitment, suggesting that the third and fourth year might shape commitment to future work [27,30]. An exemplary education program effectively increases all dimensions of professional commitment among third-year nursing students [26]. Other factors influencing the dimensions “desire to stay in the profession” and “intrinsic positive value of work” might exist, and interventions targeting them could be explored in the future.

Overall, the perceived teaching quality was positively related to professional commitment. Although there are few intervention studies to refer to, cross-sectional studies have demonstrated the descriptive finding that teaching quality is a positive indicator of professional commitment [29,30,51]. Professional commitment improvement is encouraged by the enthusiastic attitude of teachers and their ability to keep students interested, engaged, and excited about working in their chosen field [51]. Student feedback reflects preferences for pedagogy, serving as an important reference for faculty to improve teaching quality and, thus, students’ professional commitment. The correlation between students’ perception of teaching quality and professional commitment was positive but of low to moderate strength, indicating an association rather than causality. Future studies with larger samples and more rigorous designs are needed to clarify the impact of course quality on professional commitment.

### 4.4. Qualitative Feedback on the Course

The qualitative feedback on the course has implications for future development. The students were most impressed by CBL with group collaboration and clinical visits and expressed interest in extending the time spent on both the course and clinical visit modules. This suggests that the modules were well received. For most students, this was the first time they went to a healthcare institution and solved a health-related problem regarding their roles as future professional staff. CBL and clinical visits involved them in the real context of patient care, as expected when we designed the course. This authentic scenario aided them in learning what they can do to help people, with or without illness, solve health problems using their knowledge of health policy, coordinated care, and nursing humanities, which they learned in Module One.

The students suggested that the content of the cases should be deeper. We considered that first-year students have limited medical and nursing knowledge, so the CBL cases were designed to be easy to understand and contained minimal professional terminology. However, solving the cases still required students to apply critical thinking and seek relevant information. Interestingly, students tended to perceive the cases as lacking depth rather than recognizing their inherent challenges. We acknowledge and respect the students’ feedback and will continue to refine both the case content and overall CBL process in future iterations.

Nursing humanities is another type of content that impressed the students. This content was new to them and broadened their understanding of the nursing profession. Compared with other lecture content, nursing humanities are easier to comprehend and attract more interest because they require less specialized knowledge to understand, using many real scenarios. Integrating the humanities into the nursing curriculum increases the ability of nursing students to see patients holistically, enhances their understanding of themselves, increases sensitivity to the voices of others, and facilitates alternative ways of learning [52]. Nursing students must be prepared to understand the complexities of nursing practice beyond acute care and examine the influence of sociocultural determinants and their links on the health behaviors of various patient populations [53].

The students also provided suggestions for teaching pedagogy in the lecture module. First, cases are needed to help understand theoretical knowledge. Teaching pedagogy positively impacts students’ knowledge and interest in learning. The suggestion about the use of authentic cases is consistent with evidence suggesting that using clinical scenarios that integrate knowledge is helpful for first-year students to understand content and increase their satisfaction and learning motivation [39]. Additionally, more class interactions between instructors and students are expected to improve student enthusiasm and engagement. Empathetic interaction, which asks the teachers to put themselves in their students’ positions, reduces the separation and boredom of students in the learning environment during intensive interaction and leads to a deeper understanding of the processes and consequences of education [54]. Immediacy refers to teachers’ verbal and nonverbal behaviors that decrease the perceived psychological distance between teachers and students [50]. Training in teaching immediacy is suggested for faculty members to help them improve instructional effectiveness.

### 4.5. Limitations

A major limitation of the pre-post design is that it cannot prove causality, and there may be changes in the outcome variables that are not due to the course. The lack of a control group further limits our ability to attribute observed changes solely to the intervention. However, there were no concurrent nursing specialized courses in this study; therefore, we can exclude the influence of other major-related factors. In addition, quantitative and qualitative feedback on teaching quality were examined to provide a comprehensive understanding of the results. However, a longitudinal study is needed to examine how professional commitment develops over the four years of undergraduate education and whether the effect of the course on professional commitment can be maintained in the long term. Another limitation was the small sample size. Although we obtained the consent of all 50 students to participate, the effectiveness of the course on professional commitment should be examined by applying it to a larger sample at other universities. Moreover, we used convenience sampling, which may limit the generalizability of the findings. Finally, due to the small number of students in each cohort, the questionnaire used to assess the students’ perceived teaching quality was not formally validated before this study. Therefore, reliability (e.g., test–retest reliability) and validity (e.g., content and construct validity) were not established. We will test the reliability and validity of the questionnaire in the future.

## 5. Conclusions

A newly developed introductory course, incorporating sociopolitical concepts of health and advanced nursing knowledge, had a positive effect on the professional commitment of first-year undergraduate nursing students. The students had a satisfactory experience with a multi-module course that combines theory with practice. Although this study was conducted in the context of Healthy China 2030, its implications may be useful in a larger community, as every country has its own national health policy and is developing under the promotion of global health. The findings suggest that nursing educators should consider the external macro-sociopolitical environment when designing an introductory course to equip students with a broader perspective on nursing professional development. Teaching pedagogy should be improved to make students interested, engaged, and excited in the course, as well as to internalize the knowledge effectively to achieve the ultimate goal of improving professional commitment.

## Figures and Tables

**Figure 1 nursrep-15-00310-f001:**
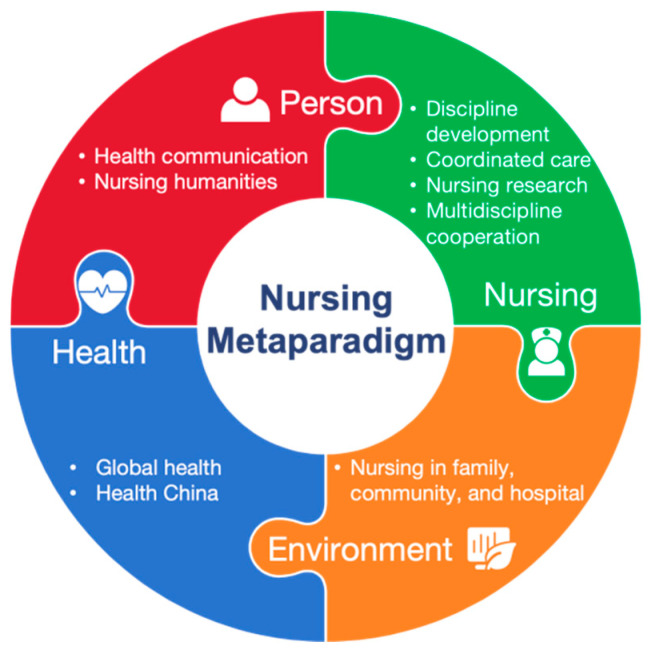
Course framework guided by the Nursing Metaparadigm.

**Figure 2 nursrep-15-00310-f002:**
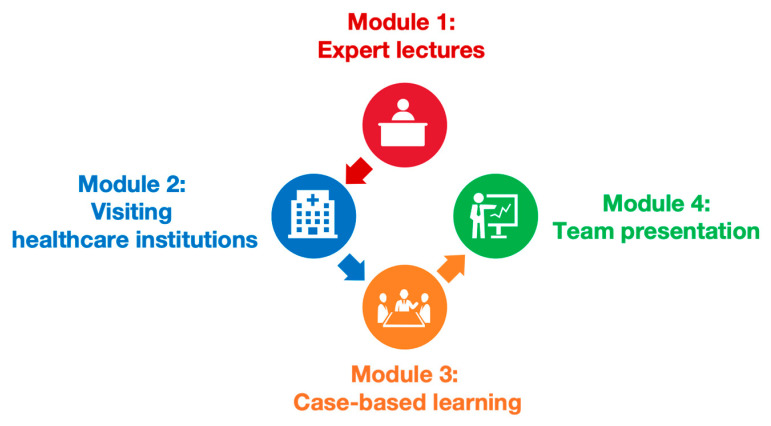
Outline of the organization of the Health and Nursing course.

**Figure 3 nursrep-15-00310-f003:**
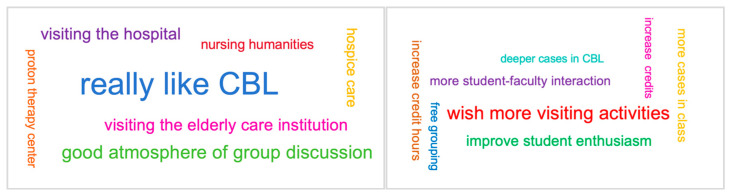
Word cloud of answers to the two questions. **Left**: answers to “What impresses you most in this course?” **Right**: answers to “What can be improved through this course?”.

**Table 1 nursrep-15-00310-t001:** Characteristics of the participating students (N_pre_ = 50).

Variable	n_pre_ (%)
Gender	
Male	14 (28.0)
Female	36 (72.0)
Residence	
Urban	40 (80.0)
Rural	10 (20.0)
Have sibling	
Yes	27 (54.0)
No	23 (46.0)
Being a student leader	
Yes	12 (24.0)
No	38 (76.0)

**Table 2 nursrep-15-00310-t002:** Students’ and their families’ cognition and attitudes regarding the nursing major (N_pre_ = 50, N_post_ = 48).

Variable	n_pre_ (%)	n_post_ (%)
Reason for majoring in nursing		
Own decision	13 (26.0)	13 (27.1)
Parents’ suggestion	11 (22.0)	10 (21.0)
Consideration for employment	9 (18.0)	9 (18.8)
Being assigned	7 (14.0)	6 (12.5)
Other reasons	10 (20.0)	10 (21.0)
Interested in the nursing major when choosing the major		
Not interested	12 (24.0)	10 (20.8)
Somewhat interested	30 (60.0)	30 (62.5)
Very interested	8 (16.0)	8 (16.7)
Parents and siblings’ current attitudes toward majoring in nursing		
Not supportive	14 (28.0)	13 (27.0)
Somewhat supportive	23 (46.0)	15 (31.3)
Very supportive	13 (26.0)	20 (41.7)
Knowledge about nursing when choosing the major		
Do not know well	24 (48.0)	22 (45.8)
Know well	26 (52.0)	26 (54.2)
Knowledge about nursing at present		
Do not know well	18 (36.0)	3 (6.2)
Know well	32 (64.0)	45 (93.8)

**Table 3 nursrep-15-00310-t003:** Comparison of the total score and the dimension scores for professional commitment per item between the pre- and post-test groups.

Variable	Before Intervention (n = 50)	After Intervention (n = 48)	*t*	*p*	Cohen’s d
WC (Mean ± SD)	2.58 ± 0.61	2.96 ± 0.82	−2.602	0.011	0.53
DC (Mean ± SD)	2.74 ± 0.59	2.75 ± 0.85	−0.050	0.961	0.02
IC (Mean ± SD)	2.79 ± 0.63	2.92 ± 0.64	−0.971	0.334	0.21
BC (Mean ± SD)	3.17 ± 0.57	3.41 ± 0.57	−2.081	0.040	0.42
PC (Mean ± SD)	2.74 ± 0.53	2.97 ± 0.60	−2.052	0.043	0.41

Note. WC: “willingness to make an effort” dimension; DC: “desire to stay in the profession” dimension; IC: “intrinsic positive value of work” dimension; BC: “belief in goals and values” dimension; PC: total score of professional commitment.

**Table 4 nursrep-15-00310-t004:** Item scores for the students’ perceived teaching quality.

	Score (Mean ± SD)
The faculty prepared for the class well, and the teaching organization was good.	3.77 ± 0.43
2. The course content was delivered clearly and easy to understand.	3.69 ± 0.55
3. The depth of course content was suitable for students.	3.63 ± 0.61
4. The proportion of class hours of lectures, visiting healthcare institutions, and CBL was appropriate.	3.62 ± 0.61
5. The content of lectures was rational and conforms to the course objectives.	3.60 ± 0.68
6. The arrangement of visiting healthcare institutions was rational and conforms to the course objectives.	3.73 ± 0.45
7. The content of the CBL was rational and conforms to the course objectives.	3.62 ± 0.64
8. Fully online teaching can achieve the same learning outcomes as the current blended course format.	3.50 ± 0.74
9. The faculty motivated and encouraged the students to think independently.	3.79 ± 0.46
10. The faculty delivered the knowledge by combining theory with practice.	3.79 ± 0.41
11. Course content and materials reflected the latest knowledge.	3.63 ± 0.61
12. Course content and materials could motivate learning enthusiasm and interest.	3.56 ± 0.74
13. The faculty provided students with adequate opportunities to ask questions.	3.71 ± 0.50
14. The faculty encouraged students to take an active part in discussions.	3.75 ± 0.49
15. The faculty responded to students’ comments and questions.	3.81 ± 0.39
16. The faculty respected students.	3.83 ± 0.38
17. The faculty delivered positive professional values to students.	3.79 ± 0.46
18. The evaluation (in-class tests, assignments, etc.) reflected the course content and objectives.	3.71 ± 0.58
19. The number of assignments and assessments is suitable.	3.67 ± 0.60
20. Online resources were helpful for understanding the course content.	3.73 ± 0.45

**Table 5 nursrep-15-00310-t005:** Pearson’s correlation analysis results between professional commitment and students’ perceived teaching quality.

	Pearson’s r to Students’ Perceived Teaching Quality	*p*
WC	0.380	0.008
DC	−0.012	0.933
IC	0.423	0.003
BC	0.062	0.676
PC	0.310	0.032

Note. WC: “willingness to make an effort” dimension; DC: “desire to stay in the profession” dimension; IC: “intrinsic positive value of work” dimension; BC: “belief in goals and values” dimension; PC: total score of professional commitment.

## Data Availability

Data sharing is not applicable to this article as no datasets were generated or analyzed during the current study.

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
