# Peer review of "Effects of an Innovative Introductory Course on the Professional Commitment of First-Year Undergraduate Nursing Students: A Quasi-Experimental Study"

_nursrep, 2025, doi:10.3390/nursrep15090310_

Round 1

Reviewer 1 Report

Comments and Suggestions for Authors

Dear authors,

The topic of the manuscript is very current and significant. It concerns the challenging issue of the impact of implementing an innovative course among first-year nursing students on their professional commitment.

Although a quasi-experimental study, it is significant because an attempt was made to collect both quantitative and qualitative feedback on the course.

However, revisions are needed so that the manuscript provides valid empirical data and new ones that will complement the existing literature on students' professional commitment.

I would like to make suggestions for improving the manuscript:

Introduction and Background

The introduction and background are coherent, easy to follow, and follow a funnel structure. The following changes are needed: Support the statements in lines 39-40 with appropriate references.

Materials and Methods

The methodology partially allows for the replication of the study.

In part 2.2. It remains unclear whether only 50 students enrol in nursing studies a year. In line 170, please enter how many students did not report, that is, were excluded from the study.

In part 2.3. clarify whether the student attended modules from 1 to 4 linearly, or if there was a possibility that they attended module 4 from module 2, as can be concluded from Figure 2. If that option was foreseen, please indicate the criteria for moving from module 2 to 4 and the further way of attending the course.

All socio-demographic data should be listed in part 2.4.1 for a clearer insight into the obtained results. Please complete this information.

For the statements in lines 257-258, it would be significant to state more specifically whether the faculty members, although not connected to the course, are involved in teaching nursing students and what the criteria were for their engagement.

The example for statistical data processing in lines 264-267 is inconsistent with the statements in part 2.4.3. I recommend that you reconcile the data.

For the tests you listed in lines 269-272, please indicate whether and with which tests you checked the normality of the sample distribution.

Results

The data from lines 252 and 275 are inconsistent and may lead to confusion, although it is clear that two students did not complete all surveys. The distribution of student responses to the statement "Whether know well about nursing when choosing the major" is particularly unclear. Please correct/clarify this.

The data in Table 2 would be easier to follow if the table were formed by transferring the data from rows to columns. Considering the importance of the study, it is essential to calculate the effect size indicator to get a clear insight into the size of the effect of the intervention, that is, the course.

In Table 3. In addition to the marks for the statistical significance level, I suggest you also list other relevant data related to the statistical method by which the significance was determined.

In part 3.4. it would be significant to expand with specific data/statements obtained from students.

Discussion

The discussion is extensive, and the authors draw attention to many important questions raised by their research and the results of other authors, through a logical discussion structure with adequate references from the literature.

Conclusions

The conclusions are concise, well-argued, and based on research results.

References

The references mentioned are relevant to the topic that the paper dealt with.

I hope you find my comments helpful.

Reviewer 2 Report

Comments and Suggestions for Authors

In the Abstract, the name of the tool being used to measure your data should be mentioned. 

Line 15- the professional commitment was compared how? using what?

Line 15- the correlation was assessed using what method? 

Line 19- the sentence that starts 'the most impressive aspects' should either be reworded, meaning is not clear since you haven't discussed the methods yet.  

Try to keep the abstract succinct and clear. It seems there are a lot of specifics mentioned that one would have to read the paper to understand first.

Introduction. The first paragraph is not clear. Needs to have a connection between the shortage and the professional commitment, then the next paragraph on global health is good, but they don't flow together.  Where is the connection between the shortage, professional commitment, and global health?

Line 32- the wording is not right, 'but also globally' doesn't fit. would sound better to say 'global healthcare'.

Line 32- need a reference for the 27 million men and women in nursing. Not a stated fact known by all.

Line 33- Meaning of the two sentences together isn't clear- reword the sentence starting with "However".

Line 35, need a better transition into your topic.  starting with a shortage is ok, but how do you get from the topic of a shortage to Professional commitment?  Maybe just add a sentence about why there is a shortage or what the data says about the shortage and how it connects to your topic of professional commitment.

Line 38-word "Letting" is not clear. Are you allowing them to do this or helping them to do this? If it is intentional and work that you are doing, then letting is not the right word.

Line 50- try to keep language neutral and not biased. When you say ' breakthrough' you are putting positive connotations on an initiative.

Line 56- it shouldn't say 'implies' that would mean there is sub context in the mandate, reword.

Line 59- should say 'rather than only focusing on patient care' if you don't put in only, then you are saying don't focus on patient care.

Line 60- wording change for 'against this background" - it doesn't make sense. Could just start with 'The Health and Nursing course'

Line 64- sentence is confusing.

Line 66- do you mean 'Healthy China 2030' or do you mean China in a state of being healthy?

Line 73 and 74, do you have data to back this up or is this an assumption? May want to be more vague and just say that Traditionally, the first course nursing students take focuses on the history, basic concepts, ... OR, just point out what is different about the newly developed class.

Line 80- Change the word 'letting' to something more descriptive of what you are doing in class.

Line 87- delete "designed as the first specialized course', you already said this in the previous paragraph.

Line 88- should say 'the' nursing metaparadigm.

Line 90 and 91-delete biased words like 'extraordinary' and condense sentence to not reiterate what was said in the previous sentence.

Line 92- take out personal language. Say, 'course content was framed around..."

Figure one- seems very similar if not exact to the current figures of the Nursing Metaparadigm- should cite this.

Line 108- take out personal reference, just say "previous study suggested...."

Line 129- take out 'to the best of our knowledge'

Paragraph starting with line 129. It is unclear what you are focusing on here if it is third- or fourth year, internship, higher grades verses lower grades (does this mean grades awarded for classes or grade levels and how does that compare to the nursing year?)

Line 148- what types of books are there that aren't human? I don't know what this means.

Line 167- how many of the cohort chose to take the pre and post test? was there any benefit?

2.2 ethical consideration- were any of the instructors of the course the research investigators or were they from outside of the course?

Figure 2- if the modules were sequential, why doe module one point to both module 2 and module 4? This diagram is confusing.

Line 187- should say Healthy China

Line 195- wording doesn't quite make sense.

Line 196- no comma needed, also sentence is too wordy, doesn't make a lot of sense.

Line 201- it was organized twice? What does that mean? Does it mean that there were two class periods or that there was two learning modules? Unclear

Line 212- unclear meaning

2.4.1 if you are gathering data other than demographics- how did you determine that the questions were reliable?  Was there a pilot done or did you discuss with experts in the field?

Line 225- if you are going to say widely used, then where is it used? how many times?

2.4.3. Again, how was the questionnaire tested for reliability? Where was the development?

2.4.4 how do these open ended questions relate to your goals of either improving the professional commitment or evaluating perceived teaching qualities?

2.5- how were these things assessed? Be clear about which tools you used at this point.

2.6 your mean and SD weren't used to describe but to define

Line 262- 'numbers and percentages'? This is too vague and not sure what you mean. Also, don't analyze your data in this section, just describe how you analyzed it. The results and analysis sections are where you talk about your n1 and what to improve.

Results- what does it mean when it says 'half the students knew about their major when choosing it'.? Did half not know what nursing was?  Why would they choose that major then?

Line 281- was the only different factor for the two students who didn't complete the post -intervention survey that they weren't interested in nursing?  Or was it that they were female?  I wouldn't put the data like this.

line 287- should be a space between paragraph and table

Table 1- Much of this I wouldn't publish as it is just highlighting where the two who didn't take the post survey scored and that could be more of an identifier. Starting with the Being interested in the nursing major- I would have a table between the pre and post.  The data above that could be in a table alone as demographics (which shouldn't change between pre and post tests).

3.3- where did this questionnaire come from? Is online teaching not considered teaching?

3.4 the qualitative feedback does not seem significant to the aims of the study. This data is not well developed or described as to how things were grouped or what was important to what extent. This section should be rewritten.

4.2 section is not backed up with data beyond the study and if questions were self created... then the relevance is negligible.

4.4- no quotes used to show data- this is a detriment to your reporting. If you want the reader to understand the data, you need to include exemplars

Paragraph starting line 444- not clear, also no direct quotes.

line 490- innovative is a word with connotations, replace

Tense of verbs changes, many times past tense and then current tense- these should be standardized.

Line 436- more credit hours may mean that they think they should get more credit, not that they wanted more time in it. 

Reviewer 3 Report

Comments and Suggestions for Authors

Dear Authors,

Thank you for submitting your manuscript. Please refer to the attached file for details. I hope that my comments will contribute, even in a small way, to the improvement of your manuscript.

Best regards,

Reviewer 4 Report

Comments and Suggestions for Authors

Dear authors,

I find this topic interesting because analyzing the relationship between nursing students' professional commitment and the quality of the teaching received would allow for the implementation of measures to improve teaching and foster the commitment of nursing professionals, improving the quality of patient care.

The introduction adequately addresses aspects related to nursing student commitment.

The objectives are clearly defined.

The methodology is explained in detail, allowing for the study to be repeated, although it should be indicated that this is a quasi-experimental study without a control group (Line 162).

Were the 50 participants all first-year students? Were there any who did not participate voluntarily?

The type of sample used should also be indicated: non-randomized convenience sampling.

Do lines 236-237 correspond to the study by Lu et al. (2002) or to this study? If they pertain to this study, they should be included in the results section. The same applies to lines 244-246.

They indicate that a questionnaire was used to assess the quality of teaching perceived by students. I believe the process followed for its implementation, its reliability (not only internal consistency, but also homogeneity and test-retest reliability), and its validity (content, criterion, construct) should be discussed. If the questionnaire has not been validated, the significant limitation this poses to the validity of the results obtained should be indicated.

The average used to complete the questionnaires should also be included in the results section (lines 256-257).

Likert-type scales generate ordinal qualitative variables, so I believe independent samples t-tests and Pearson's test would be appropriate for quantitative variables; other statistical tests should be chosen.

It should indicate in detail how the level of perceived teaching quality will be assessed.

The results are presented in tables that facilitate understanding.

I believe that, in addition to the mean and standard deviation, symmetry should be added to determine whether the questionnaires used are sufficiently discriminatory and representative of the study population.

I believe that a table should be included that statistically relates commitment to teaching.

I believe that the results of the qualitative research should be more detailed, indicating the categories into which the responses to the questions are grouped. If possible, a table should be created to facilitate the reader's understanding.

The discussion addresses in detail and by section all the aspects analyzed in this study, analyzes the results in depth, and establishes relationships with previous studies on the research topic.

Limitations should include those arising from the lack of a control group, the type of sampling, and the use of a non-validated questionnaire.

The conclusions reflect the proposed objectives.

The references are adequate, although some should be updated if possible.

Kind regards.

Round 2

Reviewer 2 Report

Comments and Suggestions for Authors

This looks good.

Reviewer 3 Report

Comments and Suggestions for Authors

Dear Authors,

Thank you for revising and adding content based on the reviewer comments within such a short period of time. I have confirmed that the comments have been appropriately addressed and reflected in the manuscript.

Best regards,

Reviewer 4 Report

Comments and Suggestions for Authors

Dear Authors,

I consider that the manuscript has been sufficiently improved and I have no additional comments or suggestions.

Kind regards.